# Nutrition, Obesity and Asthma Inception in Children. The Role of Lung Function

**DOI:** 10.3390/nu13113837

**Published:** 2021-10-28

**Authors:** Sanchez-Solís Manuel, García-Marcos Luis

**Affiliations:** 1Paediatric Allergy and Pulmonology Units, Virgen de la Arrixaca University Children’s Hospital, University of Murcia, El Palmar, 30120 Murcia, Spain; msolis@um.es; 2IMIB Bio-health Research Institute, El Palmar, 30120 Murcia, Spain; 3ARADyAL Allergy Network, El Palmar, 30120 Murcia, Spain

**Keywords:** asthma, obesity, epidemiology, dysanapsis, childhood

## Abstract

Obesity is an important public health problem. WHO estimates that about 39 million children younger than 5 years of age are overweighted or obese. On the other hand, asthma is the most prevalent chronic disease in childhood, and thus, many children share those two conditions. In the present paper we review the epidemiology of children with asthma and obesity, as well as the consequences of being obese on the respiratory system. On the one hand obesity produces an underlying T-helper 2 (TH2) low inflammation state in which numerous cytokines, which could have an impact in the respiratory system play, a role. On the other hand, some respiratory changes have been described in obese children and, specially, the development of the so called “dysanapsis” (the disproportionate scaling of airway dimensions to lung volume) which seems to be common during the first stages of life, probably related to the early development of this condition. Finally, this review deals with the role of adipokines and insulin resistance in the inception and worsening of asthma in the obese child.

## 1. Introduction

Obesity has become a real pandemic and is a major public health concern; the World Health Organization estimates that in 2020 there were 39 million overweight or obese children under the age of five years, and that in 2016 there were over 340 million overweight or obese children and teenagers aged 5–19. The WHO additionally stated that there had been an almost threefold increase in worldwide obesity since 1975 [1], although in some countries that increasing trend has levelled off in the last 10 years [2]. The high prevalence of overweight and obesity is observed at young ages, such as from 2–7 years old; overall, the pooled prevalence estimate regarding overweight/obesity in European children (aged 2–7 years) for the years 2006–2016 was 17.9% (95% CI: 15.8–20.0), whilst the pooled prevalence estimate of obesity was 5.3% (95% CI: 4.5–6.1) [3]. It is also important to recall that there is a clear correlation between childhood or teenage BMI and adult BMI [2] and, so it can be stated that it is very probable that the obese child or teenager continues to be obese at adult age; thus the consequences of childhood obesity on public health will remain for many years.

On the other hand, asthma is the most frequent chronic childhood disease: the mean worldwide symptom prevalence of current wheezing in the last 12 months is about 13% in adolescents (13–14 years) and 11% in children (6–7 years) [4]. It is thus not uncommon for these two prevalent diseases to coincide in a considerable number of children; if the prevalence remained the same as in the general population there should be approximately 2% of obese, asthmatic children. If asthma favoured the development of obesity or vice versa, then this figure would have to be even higher.

Numerous cross-sectional and cohort studies have shown an association between obesity or overweight and asthma [5,6,7,8,9,10,11,12,13,14]. Longitudinal studies have shown an increase in the risk of persistent wheeze among those children who are within the uppermost quintile of increase in BMI in the last year with respect to those children who are within the third quintile [10] and the proportion of children who develop asthma is greater among those who stayed in the 85 percentile or higher of BMI during the follow-up of around 14 years [15]. However, one very broad study that included 16 European cohorts found that it was asthma that increased the risk of obesity (hazard ratio (HR) 1.87 (95% CI 1.32; 2.64)) [16]. Physiopathological mechanisms can be imagined that explain both directions in this relationship; thus, asthmatic children generally carry out less exercise and receive treatments that increase their appetite and weight, so they become at risk of developing obesity; on the other hand, obesity is a process that presents a subclinical inflammation, in which there are endocrine alterations and, furthermore respiratory function changes, all of which could affect the respiratory tree, developing asthma. It is probable that this is a two-way street and, in any case, causes a worsening in the child’s quality of life [17] and asthma severity: they suffer from more hospitalisations, more admittance to intensive care units and use more medication [18]. BMI is an independent risk factor for severe asthma (odd ratio (OR) = 1.12; 95% CI: 1.05–1.21) compared to suffering from moderate asthma, mild asthma or not suffering from asthma [19]. In a total of slightly more than 100,000 hospitalisations due to asthma in children aged between two and 18 years, it was shown that obesity was a risk factor for the need for mechanical ventilation (OR = 1.59; 95% CI: 1.28–1.99) [20]. Conversely, a worse bronchodilatory response to albuterol has been shown; the risk of not having a bronchodilatory response is significantly higher among obese children and teenagers independently of age, gender, ethnicity, base lung function and prior controlling treatment (OR, 1.24; 95% CI, 1.03–1.49) [21]. Moreover, a worse response to budesonide has been described among obese patients in terms of lung function but also in the number of visits to the emergency room and hospitalisations due to asthma [22].

## 2. Consequences of Obesity on the Respiratory Apparatus

### 2.1. Inflammation

Obesity associates a non-TH2 subclinical inflammatory state induced by the infiltration of M1-type macrophages capable of liberating pro-inflammatory cytokines such as IFN-γ, IL-6, TNF-α, IL-1β and monocyte chemotactic peptide (MCP)-1 [23]. It has been shown that in obese adults with asthma, the number of M1 macrophages in visceral fat is correlated with the body mass index (BMI) [24]. Thus, obese people generally show elevated TH1-type cytokines with respect to their healthy peers, and this difference has also been shown when asthmatic obese people are compared to non-obese asthmatics [25]; for example it has been found that in asthmatic children, those who are obese, with respect to those who are not obese, have significantly higher values of IFN-γ and lower values of IL-4, which speaks in favour of this TH1 imbalance, as opposed to TH2 in the obese [26].

The classification of asthma into two phenotypes is generally accepted, from the point of view of immune response; the so-called “T2-high” whose essential characteristics are the overexpression of the immune type 2 (TH2) ways, that is to say of the IL-4 and IL-13 genes and which also shows in adults, the highest sputum eosinophilia and exhaled nitric oxide fraction, and was restricted to severe asthma that involves oral corticosteroid dependency, frequent episodes and severe airflow obstruction [27]. On the other hand, the phenotype in which the TH2 response is not over expressed is defined as “TH2-low” and whose characteristics are worse defined; although, the overexpression of genes related with the inflammasome and superfamilies of interferon (IFN) and tumour necrosis factor (TNF) has been shown and that, in adults, they usually manifest with an increase in neutrophils in sputum and not of eosinophils and, in consequence, a TH1-type response [27,28].

Thus, from the point of view of the immune response, it could be said that obesity has characteristics of the T2-low phenotype since obesity brings on a state of chronic low-grade inflammation characterised by elevated plasma concentrations of inflammatory cytokines such as IL-6, being shown that said inflammation associates strongly with disease severity in asthma [29]. Furthermore, a strong negative correlation between plasma IL-6 concentrations and expression of the CD8+ T-cell genes network, in sputum samples from obese adult patients has been found, compared to healthy controls; the authors speculate that obesity-related systemic inflammation may lead to CD8+ cytotoxic T-cell dysfunction, possibly by means of T-cell exhaustion [30].

A recent study found, on this occasion in children, new evidence of the transcriptome of CD4+ T cells derived from asthmatic obese children being different from that of CD4+ T cells from normal weight asthmatic children; specifically several genes associated with the small GTP-binding protein CDC42, which plays a role in T-cell activation, were upregulated in T cells from obese asthmatic children. Expression of genes downstream of CDC42 as MLK3 and PLD1, in the MAPK and mTOR pathways, respectively, was upregulated in TH cells from obese asthmatic patients, thereby suggesting that the pathways activated distal to CDC42 may play a role in the non-atopic TH1 immune responses observed in obese asthmatic children and, additionally, that Log10 transformed CDC42EP4 and DOCK5 gene counts were inversely correlated with the FEV1/FVC ratio, but only in obese asthmatic children [31].

In the obese patient’s adipose tissue, the majority of the macrophages that infiltrate are M1 type, which produce inflammatory cytokines such as IL1-β, IL-6 and IL-15 that act on naïve T cell receptors, inducing the differentiation on TH17 lymphocytes. These TH17 lymphocytes, activated by IL23, also secreted by M1 macrophages in turn liberate, IL-17A, IL-17F, IL-21 and IL-22. The liberation of IL-17A closes the circle because it activates M1 macrophages and dendritic cells that reinitiate the liberation of those same cytokines [32]. The IL-17 has been found to be elevated in adult asthmatic obese patients compared to overweight asthmatics and thin asthmatics, in both plasma as well as in sputum and the levels of IL-17 in sputum are correlated with the presence of neutrophils in the sputum (r = 0.353; *p* = 0.002) and with FEV1 (r = –0.47; *p* < 0.001) [33]. The hypothesis has been put forward that non-atopic asthmatic obese patients could have a hypomethylation of the THFA promoting genes in M1 macrophages and the IL-17A promoting genes in TH17 lymphocytes; the result of such a genetic overexpression would be the elevation of TNF-α and of IL-17A [34]. In children with central obesity, higher percentages of Th17 cells have been found than in children from the control group [35] and it was also shown that, after Ionomycin stimulated peripheral blood mononuclear cells, there is a significant increase in TH17 cells (34.7 ± 1.54% vs. 25.4 ± 2.38%; *p* = 0.0023) in obese children without asthma, allergic rhinitis, atopic dermatitis nor autoimmune diseases, compared to the non-obese controls. This increase correlates with BMI (r = 0.42; *p* = 0.0005) and the relative mRNA expression of RORC (*p* = 0.013) and IL-17A (*p* = 0.014) were both upregulated in the overweight children compared to those who were not overweight [36].

In summary, obesity induces a situation of persistent TH2-low inflammation which is probably initiated by M1 macrophages infiltrating adipose tissue. Those cells start then a cascade of cytokines release that, acting upon CD4 lymphocytes, maintain elevated levels of interleukins such as IFN-γ and IL-6, typical of TH2-low inflammation. The levels of those cytokines are related to neutrophiles in the airway of individuals with impaired lung function [33].

### 2.2. Changes in Lung Function

One study of almost 400 adults described a significant drop, in relation with the BMI, of the total lung capacity (TLC), expiratory reserve volume (ERV), functional residual capacity (FRC), vital capacity (VC) and residual volume (RV) [37] and, yet did not describe alterations in the forced spirometry [38]; except for the FVC which may be dimished [39].

The pattern in children seems to be different and, although there is also a fall that correlates to BMI in FRC and RV, the most relevant characteristic is the decline in the FEV1/FVC ratio without a loss in FVC, which in fact even increased [40]. A broad study involving more than 1200 respiratory function studies in non-asthmatic children between the ages of six and eight years, showed a significant increase in FEV1 and FVC with the increase in BMI, but both increments were not parallel: there was a disproportionate increase in FVC compared to FEV1, so the FEV1/FVC ratio decreases with the increase in BMI; with childhood obesity being associated with lower FEV1/FVC because of a disproportional increase of FVC with respect to FEV1 with increasing BMI [41]. Similar results have been replicated in another wide-ranging study that evaluated the effect of overweight on lung function in 1717 children aged five to 18 years; they also found a positive association between the BMI z-score and FVC and FEV1 but, conversely, a negative association with FEV1/FVC and these lung function alterations presented independently of whether the child was atopic or not [42]. The broad ISAAC study (around 10,000 children with obesity or overweight) also found this significant linear decline in the ratio of FEV1/FVC with BMI; once again because there was an increase in FVC without significant changes in FEV1. The authors concluded that there was an association of BMI with objective markers of airway obstruction (FEV1/FVC) but that there was no such association with other objective markers of allergic and respiratory disorders, such as bronchial hyperreactivity (BHR), skin prick test, or total IgE [43].

In 2018, Forno et al. [44] published a meta-analysis regarding respiratory function alterations and their relationship with obesity both in adults as well as in children and found that in adults there was a significant reduction in FEV1 (−2.38 (−2.87, −1.88)), but not so in children (−0.82 (−2.27, 0.63)); they noted a significant decline in FVC in adults (−4.56 (−6.92, −2.20)) but not in children in whom there was a slight, albeit not significant, increase (0.33 (−1.66, 2.33)); a significant reduction in the FEV1/FVC ratio was observed in adults (−1.00 (−1.44, −0.57)) and this was even greater in children (−2.41 (−2.98, −1.84)) and additionally a significant decline in TLC and RV in both groups. However, there was a clear reduction in FRC in adults (−23.17 (−30.26, −16.07)) and this was unchanged in children (−3.74 (−15.02, 7.55)). The authors [44] concluded that in that meta-analysis, regardless of asthma status, overweight/obesity was shown as being detrimental to lung function across the age groups. Moreover, such detrimental effects were not similar for adults and children: obese adults had a more pronounced decline in FEV1, FVC, TLC, and RV, as opposed to the obese children, who showed a steeper decline in FEV1/FVC and FEF25-75.

It therefore seems clear that in children obesity produces a growth in lung function dysanapsis; that is to say that during childhood growth, obesity influences the lung volume differently than on the flow as a consequence of incongruences between the growth of the parenchyma and the calibre of the airway, so the FEV1/FVC ratio declines, although the values of both FVC and FEV1 are within normal limits. Recently, it has been shown in children aged 10 years that both dysanapsis and suffering from asthma (current asthma) are statistically significantly related with the visceral fat index (measured by magnetic resonance imaging (MRI)), independently of the total fat mass (measured by dual-energy radiograph absorptiometry (DXA)) [45]. This is an important aspect since, as is described below, the adipokines leptin and adiponectin and also insulin resistance seem to play a role in the development of asthma in obese patients, and both circumstances are better related with the amount of visceral fat than with subcutaneous fat [46].

There would also appear to be a relationship of dysanapsis with the genetic regulation of the ways regulating the TH1 expression: the study by Rastogi et al. [31] identified that the CDC42 way is upregulated in the TH cells of obese asthmatic children and that the expression of the CDC42EP4 and DOCK5 genes, implicated in said way, are inversely correlated with the FVE1/FVC ratio only in obese asthmatic children, but not in non-obese asthmatic children. This suggests a role for this way in the TH1 non-atopic systemic inflammation and the changes in lung function found in patients with asthma related to obesity.

Dysanapsis has clinical consequences. The study by Forno et al. [47] included a total of 4521 youngsters aged between six and 20 years, belonging to six different international populations that included both asthmatic and non-asthmatic children. Considering dysanapsis as a z-score of FVC > 0.674 and z-score of FEV1 > −1.645 but FEV1/FVC < 80%; some 26.2% of the children presented that change. BMI was significantly associated with dysanapsis (OR = 1.44; 95% CI: 1.31–1.58) but, moreover, showed that the longitudinal increment of one BMI point is a risk factor for dysanapsis (OR = 1.93; 95% CI: 1.67–2.23); in fact, those children who had always been obese had an odds ratio of developing dysanapsis of 4.31 (95% CI: 2.99–6.22) compared to those who had always kept to normal weight. Additionally, the patients who presented dysanapsis had, as compared to those who did not present it, use of at least three medications for asthma (OR = 1.72; 95% CI: 1.02–2.90); daily albuterol use (OR = 8.3; 95% CI: 1.1–64.0); missed more school days (OR, 10.5; 95% CI: 1.9–58.5); more risk of, at least, one hospitalisation due to asthma in the last year (OR, 3.03; 95% CI: 1.10–8.37) and also that among those with overweight or obesity who had dysanapsis compared to those that did not, there was an increase in the risk of having a severe exacerbation (HR = 1.40; 95% CI: 1.07–1.75) and of needing at least two cycles of prednisone between visits in the study (HR = 2.53; 95% CI: 1.71–3.73).

Dysanapsis seems to start very early on; the variability of the mesoflow for a given lung volume among individuals remains constant between six years of age and adulthood; therefore, a child with a small airway for their lung size will retain that proportion throughout their life; that is to say that the dysanapsis has its origin in early infancy [48]. In fact, it has been described that those infants with the greatest weight gain in the first year of life are inversely associated with the change in lung function, and the authors proposed that such associations may prove relevant to the clinically-recognised syndrome of the ‘‘fat happy wheezer” [49]. On the other hand, one study that included data from almost 25,000 children belonging to 24 cohorts, showed that weight gain was inversely related with the FEV1/FVC, measured between 3.9 and 19.1 years, which suggests dysanaptic growth, and that excessive weight gain is a risk factor for asthma (OR = 1.27; 95% CI: 1.21–1.34); moreover, mediation analyses have hinted that FEV1, FEV1/FVC ratio, and FEF75 may explain 7% (95% CI, 2% to 10%) to 45% (95% CI, 15% to 81%) of the associations between asthma and early growth characteristics [50].

Conversely, obesity has also been related with increased bronchial hyperreactivity: the maximum fall in FEV1 after exercise was significantly greater among obese asthmatics than among non-obese asthmatics and that the leptin plasma levels, significantly more elevated in obese subjects, were related with the risk of exercise-induced bronchospasm (OR = 1.21; 95% CI: 0.027–1.335) [51]; the risk of exercise-induced bronchospasm is thus higher the higher the BMI is [52]. The dysanapsis plays an important role in the HRB of the obese; thus the FEF25–75/FVC ratio, which is also considered as a dysanapsis marker, is related with an increase in the bronchoconstrictor response in the cold air challenge test [53] and likewise, with the sensitivity to the methacholine test [54,55,56].

Obesity in children, which is mainly due to accumulation of visceral adipose tissue, starts a process of dysanaptic lung development characterised by decreased growth of airway diameter in relation to lung volume. This anomalous development provokes certain degree of airway obstruction which has clinical consequences and may explain in part asthma inception in children who become obese and increased severity in those who suffer from comorbidity.

## 3. The Role of Adipokines and Insulin Resistance

The possible role played by cytokines liberated by the adipose tissue has been studied, especially the role of leptin, with contradictory results. Guler et al. [57] in 102 asthmatic patients versus 33 healthy controls, found that leptin was a predictive factor for asthma (OR = 1.98; 95% CI: 1.10–3.55). However, the study by Kim et al. [58] found resistin to be a predictive factor for asthma (Log resistin OR = 0.587; 95% CI: 0.35–0.98) but not leptin (Log leptin OR = 0.94; 95% CI: 0.65–1.35) which, however, did correlate inversely with FEV1 and FEF25-75. Other authors also found no relationship between leptin and asthma [59,60]. A metanalysis published in 2017 [61] found asthma diagnosis to be associated with elevated levels of leptin in adults (standardised difference in means = 1.37, 95% CI 0.62 to 2.13, *p* < 0.001) and in children (standardised difference in means = 0.30, 95% CI 0.01 to 0.59, *p* = 0.042) and low levels of adiponectin in adults, but not in children. More recently, it has been published that leptin is not only a risk factor for asthma (OR = 1.06; 95% CI: 0.28–1.31) but that it is positively related with its severity, whilst it is inversely related with the adiponectin levels [62]. Moreover, those children from obese mothers whose cord blood leptin levels are elevated, have a greater risk of asthma at three years of age (OR = 1.30; 95% CI: 1.1–1.55) [63]. The mechanism by which the adipokines could have a role in the pathogenesis of asthma is not clearly defined; on the one hand, it has been suggested that it participates in the TH1-type inflammation since it correlates with the levels of IFN- γ and it has been described that in the presence of high leptin levels, only asthmatic obese children exhibited a TH1 polarisation with elevated IFN-γ levels and more severe asthma [26]. It may also have a role in altering the lung function [58,62]. Finally, it has recently been described that the polymorphism rs13228377 of the leptin gene is associated with higher levels of leptin in serum in asthma and those two variables (polymorphism + elevation in leptin) have a high predictive value for asthma risk (OR = 17.5, predictive accuracy 83.9%), although the authors themselves recognise the limitation of the scant sample [64]. In addition, it has been described that the single nucleotide polymorphisms (SNPs) of the leptin and adiponectin genes have a protective effect for asthma, but that that effect is lost in obese individuals [65].

A known consequence of obesity is the development of insulin resistance, and this circumstance has also been analysed in relation with asthma. Thus in 2007, a cross-sectional study of 415 obese teenagers (146 asthmatic and 269 non-asthmatic) with a mean BMI above 30 in both groups, described that insulin resistance, measured by homeostasis model assessment (HOMA), is a risk factor for asthma, with an OR of 4.1 [66]. Some years later, 21 asthmatics and 10 non-asthmatics aged between six and 17.9 years, who were not obese (mean BMI z-score: 0.15 and −0.19, respectively) were studied, and statistically significant differences were also found between both groups (HOMA: 0.7 ± 0.3 in the control group vs. 1.7 ± 1.4 in the asthmatic group; *p* < 0.01) and in addition there were 0% with HOMA-IR ≥ 1.77 in the control group and 42.8% in the asthmatic group; *p* = 0.05) [67]. In a prospective study that recruited 153 children aged between six and 15 years of age, 56 asthmatics and 97 non-asthmatics, all of whom were obese (BMI ≥ 95 percentile) the authors found that although there were no differences in the mean values of HOMA between both groups (2.25 (0.42–4.45) as opposed to 2.03 (0.28–4.97); *p* = 0.205), having a HOMA value ≥2.22 was a risk factor for allergic asthma (OR = 2.36; 95% CI: 1.01–5.49) [68]. In 2015, a cross-sectional study was published that used data from 1429 teenagers aged between 12 and 17 years who had been recruited in the National Health and Nutrition Examination Survey of 2007–2010. With this large sample, the authors found no statistically significant differences in the HOMA-IR of asthmatics and non-asthmatics (3.90 ± 0.55 vs. 3.24 ± 0.11, respectively; *p* = 0.24). However, insulin resistance was associated with worse lung function among the overweight and obese patients (FEV1: −34.32 mL (−50.99 to −17.66); *p* < 0.01 and FVC −42.28 mL (−63.22 to −21.35); *p* < 0.01). Furthermore, they showed that among the teenagers with HOMA-IR > 3.0, the increase of one point in the BMI z-score was accompanied by a loss of 70 mL in FEV1 (*p* for interaction= 0.0006) and by a significant decline in FEV1/FVC (*p* for interaction= 0.02), which did not happen in the non-insulin-resistant patients; therefore the synergic action of obesity and insulin resistance could contribute to the dysanapsis and thus to the development of asthma or, at least, to worsen it [69]. That result is thereby very interesting since it was published shortly after a study carried out on mice in which it was shown that insulin administered intranasally produced bronchial hyperreactivity and increased the depositing of collagen in the lungs [70]. However, a very broad cross-sectional study has been published recently using the data of no fewer than 11,662 children aged between three and 11 years, and of 12,179 teenagers from 12 to 19 years old. In the study, some 3703 asthmatics and 20,138 non-asthmatics were identified in the United States National Health and Nutrition Examination Survey database between 1999 and 2012. That study found no relationship between HOMA > 3 and asthma diagnosis [71], although it did not analyse the synergic action of obesity + insulin resistance that Forno et al. had found [69]. Nevertheless, all those studies are cross-sectional, so it is not possible to establish a causal relationship between obesity, insulin resistance and asthma. Cohort studies are required that analyse how the development of obesity affects the respiratory apparatus to produce asthma. Holguin et al. [72] described that, in the adult, there are two phenotypes of asthma, related with obesity, depending on the age of its onset. One group starts with asthma early on (before the age of 12 years) and presents more severe asthma, they are often atopic and there is an association between an increase in BMI and asthma duration. This suggests to the authors that in the early-onset patients, the asthma severity increases in the asthmatics who become obese whilst in those subjects who have late onset, the obesity is more likely to have a role in causing the asthma and its severity. In children it is probable that we also find both groups; some in whom obesity causes inflammatory and lung function changes and the clinical signs of asthma appear and, on the other hand, asthmatics who when they become obese, their asthma worsens. A phenotype that is independent of those related to puberty has also been described; Castro et al. [11] described, in the Tucson cohort, that girls who become obese between the ages of six and 11 years have a greater risk of developing asthma between 11 and 13 years than those whose weight remains normal (OR = 6.8; 95% CI: 2.4–19.4). Some years later, the same author related that phenotype to early menarche and defined it as that which affects obese girls whose menarche occurs under the age of 12 [73]; a phenotype confirmed recently by Chen et al. [74] who found that the synergic effect of early puberty and obesity is a risk factor for developing asthma between 12 and 17 years of age (OR = 1.08; 95% CI: 1.04–1.11).

Obesity is currently a first-degree public health problem in the whole world due to its growing incidence. Comorbidity of two very prevalent conditions such as asthma and obesity is not rare. The role of obesity on the inception of asthma and on its severity has attracted much attention during the recent years. It is reasonable to try and find common epidemiological and therapeutic targets which might be common to both diseases. We have learned that obesity is a much more complicated condition than just the collection of fat tissue and causes a state of low-grade inflammation in which insulin resistance and increased leptin levels probably play a role. Both circumstances possibly alter the developing infant lung, after provoking a non-harmonic growth of lung size and lung airway (dysanapsis) which might be the cause of a specific asthma phenotype. Nevertheless, dysanapsis has clinical consequences and would aggravate the condition in asthmatic children. Reducing the incidence of obesity would most probably reduce not only new asthma cases but also severe asthma prevalence.

## Data Availability

Not applicable.

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
