# Peer review of "Nutrition, Obesity and Asthma Inception in Children. The Role of Lung Function"

_nutrients, 2021, doi:10.3390/nu13113837_

Round 1

Reviewer 1 Report

This is a review article about the relationship between obesity and asthma in children, and the role of lung function in this regard. Howevwr, after reading it carefully, some questions arise.

  1. The writing is a bit messy, which makes it difficult to understand  some concepts.
  2. Expressions such as " have demonstrated" or "being demonstrated" (lines 46 and 99) shoul be avoided, because no study proves anything, but rather provides evidence for or against a hypothesis.
  3. In the authors´ opinion, what are the practical implications of this review? Perhaps the articleshould end  with some kind of conclusion or reflection.
  4. The references should be updated: this is a review article in  which more than 59% of the citations are older than 5 years. 

Author Response

  1. The text has been reviewed by a native speaker in order to clarify any point that could be confusing.
  2. The paper has been revised so "demonstrated" is changed for "shown".
  3. A last paragraph trying to show a summary together with the implications of the review has been added.
  4. Alhough the reviewer is right in his/her statement, a thorough search of the literature did not retrieved more that what is shown in the review. Some citations are "historical" ones which may add to the percentage of "out-of-date" but are worthy to maintain. On the other hand the scond reviewer finds that "the authors did a good revew of up-to-dtae studies. The article appears well referenced..."

Reviewer 2 Report

This manuscript provides an updated review on the relatioship of asthma and obesity  in children. The topic is relevant and practical. Moreover the authors discuss both the consequences of obesity on the lung function , inflammation and the role of adipose tissue liberated cytokines on the risk of asthma. The authors give a good review of up-to-date studies The article appears well referenced and sufficiently in depth. However, as currently written, it seems to be a bit difficult to follow and in my opinion could be more meaningful if slightly rearranged. Many individual studies are described one by one with less focus on the general message coming from metaanalyses and current recomendations. The review would gain by highlighting in more comprehensive way the conclusions at the end of each section and by adding the final message at the end of the article.

Author Response

  1. We would like to thank the reviewer about his/her comments.
  2. Following his/her suggestions and in order to clarify the flow of the reasoning behing each section, a little summary of it has been added.
  3. A final message has been also added to the text.